# Chiral Transition Metal Complexes Featuring Limonene-Derived Ligands: Roles in Catalysis and Biology

Ghaita Chahboun [1,2], Mohamed El Hllafi [1], Eva Royo [2,*] and Mohamed Amin El Amrani [1,*]

[1] Faculty of Sciences, Abdelmalek Essaadi University, Tetouan 93000, Morocco; gghaitachahboun@gmail.com (G.C.); mohamed.elhllafi98@gmail.com (M.E.H.)

[2] Departamento de Química Orgánica y Química Inorgánica, Instituto de Investigación Química "Andrés M. del Río" (IQAR), Universidad de Alcalá, 28805 Alcalá de Henares, Spain

* Correspondence: eva.royo@uah.es (E.R.); elamrani.amin@gmail.com (M.A.E.A.)

## Abstract

Chiral coordination compounds are of growing interest due to their structural diversity and wide applicability. Besides chirality, alcohol and especially oxime-functionalized limonene derivatives confer water solubility, stability, and the appropriate reactivity to enable their use in asymmetric catalysis—such as allylic substitution, alkynylation, transfer hydrogenation, and selective C–C bond formation. Biologically, they have shown promising anticancer, antibacterial, and antibiofilm activity. This review presents an integrated overview of the synthesis, properties, and applications of chiral transition metal complexes featuring ligands derived from inexpensive, naturally occurring *R*- and *S*-limonene substrates, and explore their roles in catalysis and biological activity.

**Keywords:** terpene; limonene; oxime; chiral; metal complexes; asymmetric catalysis; biological activity

## 1. Introduction

Nature is recognized as a high-class synthetic chemist, creating a vast array of chiral substrates with a high degree of stereochemical purity. Together with carbohydrates and amino acids, terpenes and their derivatives have played a significant role in modern chemistry. Terpenes are natural products built from repeating five-carbon isoprene units ($C_5H_8$) that are joined in characteristic patterns. This structural principle, known as the isoprene rule, gives rise to a wide range of hydrocarbons, which are classified as mono, sesqui-, or di-terpenes, depending on the number of assembled units. From their widespread use as flavors and fragrances to their roles as small chiral molecules with a variety of potential applications, these compounds are deeply embedded in daily life [1]. Beyond their medicinal applications [2–6], terpenes have become indispensable in contemporary chemistry as inexpensive and renewable sources of chirality, precursors in organic synthesis, biochemical messengers, potential biofuels, and building blocks for materials science [7–11]. They constitute a fundamental part of the so-called "chiral pool": nature's readily available reservoir of enantiomerically enriched molecules [12]. Despite the structural and functional diversity that makes them valuable resources for the development of chiral ligands in transition metal coordination chemistry, their potential in metal complex design remains underexplored. Furthermore, only a few reviews have addressed this topic, underscoring the need to highlight recent developments and future opportunities in this area [11,13,14].

Without intending to provide a comprehensive survey of all available data, this minireview aims to summarize those results on the synthesis, the structural properties, and

reactivity of chiral metal complexes containing limonene-derived chiral ligands, giving a special emphasis to their roles in catalysis and biological applications, and highlighting their potential in modern organic and medicinal applications.

Limonene, a chiral monoterpene that exists in nature as the two pure enantiomers (*R*)-(+)-limonene and (*S*)-(−)-limonene, serves as an excellent precursor for ligand design. Both enantiomers exhibit distinct chemical and biological properties, and the possibility of having stereoselective modification procedures available to afford pure optically active ligands able to coordinate to different transition metals open innovative applications in asymmetric synthesis and medicinal chemistry [15].

## 2. Limonene-Derived Ligands Precursors

The strategic functionalization of limonene to incorporate metal-coordinating groups preserving its enantiomeric excess is essential for the development of chiral metal complex synthesis derived from these natural products. The following section intends to highlight certain modifications of limonene specifically aimed at producing systems capable of acting as chelating ligands for transition metal complexes. A comprehensive survey of the literature related to terpenes-derived ligands was reported in 2019 by Zalevskaya et al. [14].

The addition of nitrosyl chloride to one of the two double bonds of limonene affords α-nitroso β-chloride derivatives, which can be easily transformed, stereoselectively, into α-amino β-oximes compounds through a nucleophilic substitution reaction with amines [16–22], resulting in the synthesis of a series of chiral compounds **1–19** derived from *R*- or *S*-limonene (Figures 1 and 2).

**Figure 1.** Chemical structures of limonene-derived amino-oxime ligands **1–11**.

**Figure 2.** Structures of limonene-derived diaminodioxime derivatives **12–19**.

Such amino oxime chiral derivatives can coordinate to metal centers through the nitrogen atoms of the oxime and amine functions. Notably, oxime functions are versatile frameworks that have found extensive use in medicine [23,24], catalysis, and synthesis [25,26]. Over the past 20 years it has been well recognized that a variety of molecules containing oximes are responsible for several pharmacological activities, including antimicrobial, anticancer, anti-inflammatory, antihyperglycemic, insecticidal, and antioxidant activities [23,24]. In addition, oxime derivatives serve as versatile ligands in coordination chemistry, due to their ability to coordinate through N, O, or both, leading to bidentate or bridged coordination modes [27–29]. Also, they exhibit stronger hydrogen-bonding abilities than alcohols or carboxylic acids and have attracted significant interest due to their role in directional, noncovalent intermolecular interactions. Moreover, this hydrogen-bonding capacity can enhance the solubility of the resulting metal complexes in water or alcohols [27,30].

Expanding the approach to more complex amines allowed the preparation of potential polydentate ligands, such as compounds **6** [22] and **8–11** [11,31]. Recently, Kokina et al. accomplished the synthesis of **9** and **10** (Figure 1) which have shown photoluminescent properties, both free and coordinated to metals such as Zn(II), Cd(II), and Cu(II) [31–33]. The functionalization of the oxime moiety through the reaction with 2-(bromomethyl)pyridine, led to the synthesis of the O-picolyl ether derivative **10**. The incorporation of additional donor sites not only enhanced the ligands coordination properties but also broadens the structural and functional diversity of limonene-derived frameworks, providing new opportunities in coordination chemistry.

Ethylenediamine (EDA) serves as an effective ligand in the synthesis of metal chelates. The synthesis of a variety of terpene derivatives of EDA has already been reviewed somewhere else [14]. Following an analogous procedure as that described above for α-amino oximes, Savel'eva et al. used EDA as the nucleophile to prepare tetradentate diamino dioxime **12** (Figure 2). This same reactivity allows the sequential introduction of different moieties into the EDA to produce, stereoselectively, unsymmetrical diamino dioximes such as **13** (Figure 2) [34].

Propylenediamine was used in place of EDA to introduce greater flexibility via an additional methylene group between the NH functions. This led to symmetrical derivative **14**, and to unsymmetrical **15** and **16**, combining carene or pinene with limonene-derived fragments [35]. These works also report the mononuclear corresponding complexes of Co(II), Co(III), Ni(II), and Cu(II) that were successfully synthesized from the diamine dioximes described herein. The selection of piperazine as the diamine of choice allowed to further expand the repertoire of tetradentate potential ligands to **17** [36], and analogously, the use of xylylenediamine and 4,4′- methylenedianiline leads to derivatives **18** [37] and **19** [38], reported together with their dinuclear Pd(II) compounds.

An alternative to functionalization of limonene involves the stereoselective epoxide ring opening reaction that takes place with commercial amines from *cis-* or *trans*-limonene oxide, to yield a series of β-amino alcohols, **20–30** (Figure 3) [39,40]. Additionally, the ring opening of *trans*-(*R*)-limonene oxide using lithium diphenylphosphide led to the formation of the phosphan-yl alcohol compound **31** (Figure 3) [41].

Direct N-tosylaziridination of naturally available (*R*)-(+)-limonene was employed in the diastereoselective synthesis of monotosylated diamine compounds **32–34, 32a–34a** (Figure 4) [42,43]. Notably, a tridentate N,N,N coordinating compound (compound **35**) was synthesized by Kokina et al. and demonstrated interesting luminescent properties [32].

**Figure 3.** Alcohol-amino and -phospane compounds prepared from limonene oxide.

**Figure 4.** Monotosylated diamine compounds **32–35**.

## 3. Transition Metal Complexes Containing Limonene-Derived Ligands

### 3.1. Titanium Compounds

Kuchin et al. were the first to evaluate the ability of titanium to coordinate to a variety of chiral terpene-derived alcohols [44]; the approach was later extended to the in situ preparation of Ti(IV) catalysts active in the asymmetric oxidation of sulfides [45]. Later on, further examples of Ti(IV) with α-pinene and camphor-modified ligands appeared in the literature [46–48].

To the best of our knowledge, the first examples of limonene-derived ligands coordinated to a group 4 metal was the family of mono and bis-cyclopentadienyl enantiopure α-phenylamino, benzylamino, and 2-picolylamino oximato compounds of Ti(IV) reported in 2016 and 2018 [49,50]. The crystal structure analyses of two of them demonstrated the preferential binding of the oximato group to the hard acidic Ti(IV) center of **36** (Figure 5) via dihapto kN,kO coordination. Metallic oximatos exhibit the tendency to change their coordination mode from side-on to end-on or bridging, depending on the steric demand and chemical composition of the central metal [48,51–54]. Thus, the high steric demand of the limonene residual moiety forces, in compound **37** and **37′**, to a monodentate kO coordination of the oxime, representing the first example of the kind with a unique terminal monohapto TikO coordination [50]. This type of coordination could explain the rapid hydrolysis observed for compounds **37** and **37′** in aqueous solution. In contrast, dihapto titanocene oximato complexes of the type $[(\eta^5\text{-}C_5H_5)_2Ti(H_2O)(\kappa^2\text{-}O=NR)]^+$ (R = CMe$_2$; C$_6$H$_{10}$), described by Thewalt and co-workers, were reported to be stable in both air and water [51].

**Figure 5.** Mono and bis-cyclopentadienyl Ti(IV) compounds with limonene-derived oximato ligands.

The absolute configuration of the compounds **36**, **37,** and **37′** was also confirmed through their X-ray structure. The synthesis with corresponding *R*- and *S*-limonene-derived amino oximes allowed access to enantiopure bis-cyclopentadienyl titanium compounds. The studies on their behavior in $D_2O$ at different pD values were also reported. These cyclopentadienyl Ti(IV) compounds were rapidly hydrolyzed in water, affording corresponding free ammonium-oxime derivative and cationic cyclopentadienyl Ti aquo derivatives [49,50].

De la cueva et al. [55] also reported the synthesis of a novel Ti/Pd heterometallic derivative **38** (Figure 5), from the reaction of **37** and [Pd(COD)Cl$_2$]. The coordination of the amino group to the palladium center could result in the formation of different stereoisomers but the characterization data indicated the presence of only one diastereomer in solution. Such stereoselective coordination of chiral chelating ligands derived from natural terpenes has been previously reported, this ability being one of the most interesting features found in their related chemistry [20,56,57].

Since the groundbreaking introduction of cisplatin (cis-diaminedichloroplatinum(II)) by Barnett Rosenberg in the 1960s as the first inorganic chemotherapeutic agent, there has been sustained interest in the development of metal-based drugs with improved therapeutic profiles and reduced side effects [58,59]. Titanium (IV) compounds were the first non-platinum-based metallodrugs to enter clinical trials for cancer treatment [58] and some modified titanocenes have shown great therapeutic promise [60,61]. For these reasons, this new family of titanium(IV) complexes were investigated for their potential anticancer activity, demonstrating promising cytotoxic effects against prostate cancer cell lines PC-3 and DU-145; some of them, especially **37**, outperformed the successful Tacke's Titanocene-Y (bis-[(p-methoxybenzyl)cyclopentadienyl]titanium(IV) dichloride) [60]. Despite undergoing hydrolysis under physiological conditions, the complexes exhibited synergistic effects when compared to equivalent doses of titanocene dichloride and the corresponding free amino-oxime ligands. Evaluation of isolated enantiomers revealed no marked differences in cytotoxicity across the tested cancer cell lines, with the exception of PC-3. The bis(cyclopentadienyl) derivatives were apoptotic in Caki-1 cancer cell lines, and remarkably, they affect cell adhesion and migration processes of the PC-3 cell line. They also demonstrated a degree of selectivity against non-malignant RWPE-1 prostate cells. Investigations into their DNA-binding behavior, using equilibrium dialysis, FRET melting assays, and viscometry, indicated that both the parent complexes and their hydrolysis products associate with DNA through minor groove binding or external interactions. These interactions appear to be independent of the ligand's chirality and exhibit minimal affinity at physiological pH.

### 3.2. Iron and Ruthenium Compounds

A variety of chiral Fe and Ru derivatives containing terpene-modified ligands are known [14]. More recently, stereogenic-at-metal compounds have also been prepared from pinene-derived pyridyl triazole ligand [62] and others terpene-fused terpyridines [63],

and found interesting applications in asymmetric catalysis such as the enantioselective nitrene-mediated ring closing amidation.

Regarding limonene-derived systems, a tricarbonyl limonene Fe(0) derivative **38** was prepared in 1984 (Figure 6), [64] which was later the entry to prepare a variety of carbene Fe complexes that exhibit a rich reactivity [65]. The limonene ligand is coordinated to the $Fe(CO)_3$ moiety of these compounds through both nonconjugated cyclic and acyclic double bonds.

**Figure 6.** Synthesis of Fe- and Ru-carbonyl complexes containing limonene-derived amino-oxime **2**.

In 2009, Kirin and co-workers reported the synthesis of carbonyl cluster Ru(II) complexes via the reaction of α-amino oxime **2** with the metal precursor $Ru_3(CO)_{12}$. At elevated temperatures, this reaction yields the binuclear compound **39** (Figure 6) as a 2:1 mixture of diastereomers, representing an example of *κN:κO* bridge coordination by the oxime unit. When the reaction is carried out at room temperature in the presence of a base, both bi- and trinuclear complexes **39** and **40** are formed [66]. X-ray diffraction analysis of compound **39** confirmed a "head-to-tail" bridging mode of oxime coordination. The observed Ru–Ru and N–O bond lengths were consistent with those reported in structurally related compounds featuring similar ligand coordination modes [67].

A series of α-amino-oxime ruthenium derivatives **41–46** based on limonene were actively investigated by us [22,30,68,69]. The reaction of enantiomerically pure α-amino oximes **4**, **5** (and their corresponding enantiomers), and **6** with the $[RuCl_2(p\text{-cymene})]_2$ precursor leads to the diastereoselective formation of chiral-at-metal derivatives **41**, **42** and **42′,** and **43** [68] (Figure 7). Suitable crystals for X-ray analysis of the cationic complex **42′** confirmed the configuration *SRuRN*-(2S,5R) and the kN coordination of the oxime group in this piano-stool structure with pseudo-octahedral coordination around the metal center. Remarkably, these cationic Ru(II) compounds are configurationally stable, and do not suffer epimerization processes in solution under the conditions tested [30]. In contrast with the syntheses of **41** and **42**, compound **43** was obtained as a mixture of two diastereomers in a 2:1 molar ratio, showing that the tricoordination potential of derivative **6** was important in diastereoselective control. Interestingly, derivatives **41**, **42** and **42′**, and **43** have demonstrated solubility and stability in $D_2O$ for weeks at room temperature.

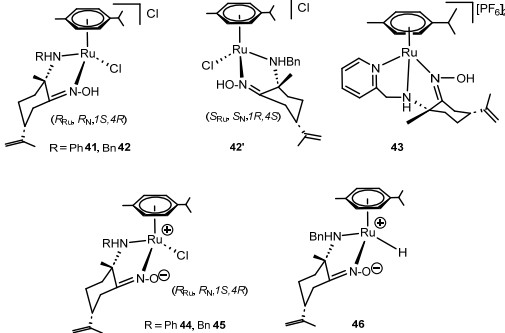

**Figure 7.** Limonene-derived ligands coordinated to arene ruthenium compounds.

The oxime moiety of these compounds can be deprotonated with KOH, allowing the formation of zwitterionic compounds **44** and **45** (Figure 7). Their molecular structures were established by X-ray diffraction [22], revealing a "piano-stool" arrangement, and

confirming a kN coordination of the oxime unit. These derivatives, together with the Ru-hydrido derivative **46** (Figure 7) were detected as intermediates during the transfer hydrogenation process of acetophenone in isopropanol catalyzed by Ru compounds **41** and **42** formed in situ [11,22] (Figure 8).

**Figure 8.** Transfer hydrogenation of acetophenone catalyzed by Ru(II) derivatives. L* is a chiral amino oxime ligand.

Chiral ruthenium compounds have been extensively studied for their excellent catalytic properties in the asymmetric transfer hydrogenation (ATH) catalysis of a variety of substrates [70–72]. Nobel laureate Prof. Noyori developed a highly active catalytic system with amine chiral ligands, and it is well recognized that the Ru–H/–NH motif plays a crucial role in generating highly catalytic hydrogenation reactivities [73,74].

First examples of the use of terpene-based ligands as chiral auxiliaries for ATH appeared in 1981 [75] and have been reviewed by Zalevskaya et al. [14]. β-amino alcohol derivatives proved to be one of the most effective chiral auxiliaries for asymmetric transfer hydrogenations catalyzed by transition metal compounds. Watts et al. showed, already in 2006, [40,76] that limonene-based β-amino alcohols can be used as ligands for catalytic ATH. Within this context, catalysts prepared in situ from proligands **20–24** of Figure 3 and arene Ru(II) precursors proved to be active in the process, especially the catalyst formed from compound **20** resulted to be highly effective in the asymmetric hydrogenation of acetophenone, with conversions of 99% and 50% ee.

In accordance with those previous results, the α-amino oxime derivatives **3–6** were evaluated in the ruthenium-catalyzed transfer hydrogenation of acetophenone by Ibn El Alami et al. [11,22], using a variety of [RuCl₂(arene)]₂ precursors in situ. All systems proved to be active in the asymmetric transfer hydrogenation of acetophenone. The best activity was achieved by [RuCl₂(benzene)]₂ and the 2-picolylamino-oxime **6** (Ru compound **41**), reaching a 94% conversion of the acetophenone. The highest enantioselectivity (80% ee) was observed with ligand **5**, (Ru precursor **42**) with a 92% conversion. The authors also contributed to the elucidation of the mechanism followed by these compounds.

The same year, Roszkowski and co-workers reported the asymmetric transfer hydrogenation of aromatic ketones catalyzed by chiral Ru(II) complexes prepared in situ by mixing [RuCl₂(benzene)]₂ or [RuCl₂(p-cymene)]₂ with (+)-(R)-limonene-derived tosylamines **32–34** and **32a–34a** (Figure 4) [42,43]. Conversion and enantioselectivity were found to be highly influenced by the nature of the ligands and of the arene moiety of Ru(II) precursors. The best results of the reduction in aromatic ketones were obtained using catalysts derived from ligands **32** and **33**. Additionally, a similar asymmetric hydrogen transfer protocol using **32–34** as pro-ligands was performed for selected endocyclic imines, where good to excellent results were obtained.

Among other metals, ruthenium complexes emerged as successful anticancer candidates, leading to the synthesis and extensive evaluation of numerous Ru-based systems for their cytotoxic properties against a broad spectrum of cancer models [77–80]. In this context, compounds **41–43** were subjected to biological testing, revealing that **42** and its enantiomer **42′** exhibited better cytotoxic activities than cisplatin in prostate cancer PC-3 cells after short times of incubation. Evaluation of compound–DNA interactions revealed that these Ru complexes associate with double-stranded DNA through external electrostatic forces and/or groove binding. Also, **42** was able to modulate the metastatic phenotype of

PC-3 cells. In xenograft studies, compound **42** achieved a 45% reduction in tumor growth and was associated with the downregulation of proteins involved in angiogenesis and cell motility in prostate tumors [30,68].

The growing demand for new antimicrobial agents to combat antibiotic-resistant bacteria has sparked considerable interest in terpene-based compounds, valued for their bio-based origin, sustainability, and innovative potential [5,6]. Limonene has demonstrated antimicrobial activity, making it an attractive ligand for metal-based drug design [81].

Khelissa et al. [82] investigated the antibacterial and antibiofilm properties of compound **42** in both its free and maltodextrin microencapsulated forms, comparing its performance to that of enantiopure (R)-limonene. The complex demonstrated strong activity against biofilms formed by major foodborne pathogens—*Escherichia coli*, *Staphylococcus aureus*, *Enterococcus faecalis*, and *Listeria monocytogenes*—exhibiting significantly greater antimicrobial efficacy than free amino oxime derivative and limonene. In particular, the minimum inhibitory concentration (MIC) of **42** (0.4 mg/mL) was approximately 30 times lower than that of limonene (12.5 mg/mL). The microencapsulated form of **42** has shown to be a promising alternative to conventional antimicrobial agents [6], showing enhanced antibiofilm activity, with up to a 90% reduction in biomass, especially against *E. coli*, *S. aureus*, and *L. monocytogenes*. Additionally, it caused more pronounced membrane disruption and $K^+$ leakage in *E. coli* than the free form.

### 3.3. Cobalt Compounds

The incorporation of terpene-based ligands to cobalt complexes have allowed the synthesis of paramagnetic Co(II) [83] and diamagnetic Co(III) complexes [35] through reactions of corresponding cobalt salts with chiral diaminodioxime derivatives derived from chiral derivatives of natural terpenes, such as pinene, carene, and limonene (compound **47**, Figure 9).

**Figure 9.** Co(III) complex with optically active diaminodioxime systems derived from limonene.

X-ray diffraction analysis of the crystal structure of complex **47** demonstrated that complexation induced two new stereogenic centers at the amino N atoms, leading to three possible diastereomeric forms (X, Y, and Z) (Figure 9) for the symmetric ligand, where only the C1 symmetry is formed, which is, again, an example of diastereoselectivity facilitated by the use of chiral terpene-derived ligands.

### 3.4. Nickel, Palladium, and Platinum Compounds

3.4.1. Nickel

Larionov pioneered the synthesis of Ni(II) with optically active ligands derived from natural terpenes, including that from limonene, derivative **48** (Figure 10) [34]. X-ray diffraction studies confirmed the formation of ionic complexes of paramagnetic nature with a distorted octahedral $N_4O_2$ coordination.

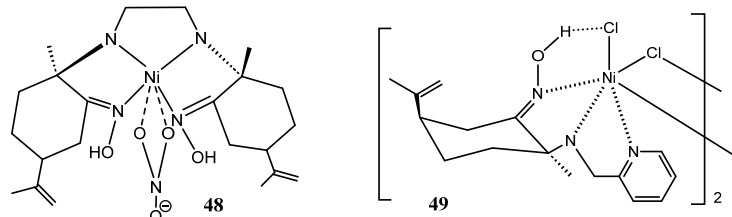

**Figure 10.** Chiral α-amino oxime nickel(II) complexes derived from limonene.

Recently, the Sauthier group designed and reported the structure of the first dinuclear Ni(II) complex with limonene-derived α-amino oxime ligands [84]. In this complex, each Ni(II) ion adopts a distorted octahedral geometry, coordinated by the three nitrogen atoms of proligand **6**, a terminal chloride, and two bridging chlorides. The Ni–N and Ni–Cl bond lengths are consistent with already reported values [85], with a Ni–Ni distance of 3.5198 Å. The crystal structure is stabilized by extensive hydrogen bonding, in a three-dimensional network.

3.4.2. Palladium and Platinum

Palladium plays a prominent role in organometallic chemistry, providing a crucial catalytic platform for a broad range of reactions, such as the Heck, Suzuki and Suzuki-Miyaura cross-couplings [86,87], Tsuji-Trost allylations, and C-H activations [88–91]. These processes enable the selective formation of carbon–carbon bonds and are essential in the synthesis of complex organic compounds. Additionally, palladium versatility extends to potential applications in biologically relevant contexts [92–94], further emphasizing its significance in various scientific domains.

Most likely, chiral palladium compounds containing terpene-derived ligands are the most abundant among all the transition metals compounds prepared [13,14]. Back in 1970, Dunne and McOuillin were the first to synthesize and characterize a variety of π-allyl Pd(II) derivatives of a variety of non-modified monoterpenes [95]. Years later, Braunstein used the enantiopure limonene-derived phosphan-yl alcohol **31** to prepare a novel family of Pd(II) compounds **50–51** (Figure 11), showcasing different coordination behaviors and stereochemical properties [96].

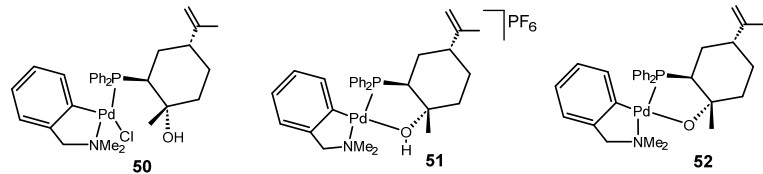

**Figure 11.** Novel palladium(II) complexes with a limonene-derived phosphino alcohol ligand.

Larionov research team [36–38] synthesized dinuclear palladium complexes **53–56** (Figure 12) using derivatives **12**, **14**, **18**, and **19** (Figure 2) as ligands. The structures of these enantiomerically pure complexes were confirmed through elemental analysis and X-Ray analysis; however, in solution, more than one stereoisomer was observed by NMR spectroscopy in different ratios, due to the epimers arising from the different configuration of the stereogenic amino N. A common feature of this dichloro Pd(II) compound is the

presence of strong hydrogen bonds between the oxime proton and one of the chloro ligands. This characteristic is observed in most of the oxime Pd and Pt chloro compounds structures known [55,97–99], due to the strong ability of the oxime functionality to produce hydrogen bonds interactions [24,27].

**Figure 12.** Mono- and dinuclear Pd (and Pt) compounds with limonene-derived amino oxime ligands.

A variety of mononuclear palladium and platinum chloro $\alpha$-amino oxime derivatives have also been reported, and their anticancer and catalytic properties in the allylation of diketones have been evaluated [30,55,97,98,100]. The bidentate potential of derivatives **4** and **5** leads to compounds **57–60** obtained as a mixture of stereoisomers. When the corresponding enantiomers **4′** and **5′**, formed from S-limonene, were used, corresponding Pd(II) and Pt(II) pure enantiomers **57′–60′** were also obtained. X-ray diffraction studies of all of them confirmed their absolute configurations. When the reaction between the amino oxime systems and the Pd precursors is carried out in a 2:1 molar ratio, compounds **61**, **61′** and **62**, **62′** were formed. In these derivatives, one of the two oxime moieties was deprotonated, forming a NO-H-ON bridge, which was confirmed also by X-ray diffraction analysis. In contrast with the stereoisomer mixture obtained in previously mentioned compounds, reaction of **6** or corresponding **6′** with the metal precursors $K_2MCl_4$ or $M(COD)Cl_2$ (M = Pd, Pt) allowed the stereoselective formation of pure enantiomers **63, 64** or **63′, 64′**, respectively, in which the 2-picolyl amino oxime derivative acts as a tricoordinated ligand.

Homrani et al. [98] explored the use of in situ generated catalytic systems derived from palladium precursors and ligands **4**, **5,** and **6** (corresponding to derivatives **57**, **58,** and **63**) in the metal-catalyzed allylation of various $\beta$-diketones and $\beta$-ketoesters (Figure 13). Initial studies focused on the reaction between dimethyl malonate and allyl acetate in the presence of the Pd/L catalyst system, predominantly afforded the monoallylated malonate, with only minor formation of the bis-allylated byproduct. Among the ligands evaluated, ligand **6** exhibited the highest catalytic efficiency, achieving up to 88% conversion and 69% chemoselectivity for the monoallylated product. The catalytic system also demonstrated excellent performance with other substrates, including $\beta$-diketones (100% conversion, 72% isolated yield) and $\beta$-ketoesters (up to 75% yield). Notably, ketoamides underwent allylation with high efficiency (86–100% conversion) and chemoselectivity toward the monoallylated products.

**Figure 13.** Pd-catalyzed allylation of β-diketones and/or -ketoesters. Dash circles means cyclic or lineal ketones.

Considerable attention has been directed toward Pd(II) complexes as promising candidates to replace cisplatin in cancer therapy [93,101,102]. Within this context, all compounds **57** to **64** (and **57′** to **64′**) were evaluated as potential anticancer agents in vitro against human prostatic cancer PC-3, breast adenocarcinoma MCF-7, and cervical carcinoma HeLa cells. Pd compounds were always more cytotoxic than Pt(II) counterparts, and in general, chloro and dichloro compounds had low cytotoxic profiles. The Pd(II) oxime-containing complexes demonstrated a notable anticancer activity, with compounds **61** and particularly **61′** showing superior potency—up to 80-fold greater than cisplatin against PC-3 cells. Compound **61′** also displayed marked tumor selectivity (selectivity index $\approx 30$) and enantiomer-dependent differences in cytotoxicity [55]. Remarkably, derivatives **63**, **64** and **63′**, and **64′** were able to modulate both adhesion and migration of PC-3 cells at concentrations that did not significantly affect cell viability [97]. A range of techniques were also applied to investigate compound–DNA interactions, leading to the conclusion that the complexes associate with double-stranded DNA through partial, non-classical intercalation, and/or groove binding, with palladium compounds exhibiting the strongest DNA-binding affinity.

The conjugation of a bioactive drug to a tumor-targeting biomolecule is a well-known strategy in medicinal chemistry to gain selectivity and reduce side effects. Vitamin B12 (cyanocobalamin) is highly demanded by tumor cells, making it a valuable "Trojan horse" for targeted delivery of metal-based drugs and imaging agents in cancer therapy [103–105]. Thus, Pt(II) derivatives **58** and **58′** were conjugated to vitamin $B_{12}$ and rhodamine-based scaffold to yield fluorescent derivatives, which were evaluated as anticancer drugs in vitro [106].

### 3.5. Copper Compounds

First limonene-derived amino-oxime Cu(II) compounds were prepared in 2002 by Larionov et al. [107]. $CuCl_2 \cdot 4H_2O$ reacted with compound **8** (Figure 1) in a 1:1 ratio to form dinuclear chloro bridged compound **65** (Figure 14). Oxime kN,kO bridged derivative **66** was obtained instead when the ligand to metal ratio used was 2:1. Single-crystal X-ray diffraction study confirms the centrosymmetric structure of **65**, with each copper in a distorted square–pyramidal geometry.

**Figure 14.** Reported copper compounds with a variety of amino-oxime ligands.

Additionally, mononuclear cationic derivatives **67** and **68** are formed when propyl bridged derivative **14** is used [108]. Increasing the length of the linker with the methylene-dianiline compound **19** permits to obtain dinuclear compound **69**, in a stereoselective manner [38].

A rare zwitterionic, dinuclear structure of Cu(II), **70**, has also been reported from the reaction of a bis-amino oxime derivative linked by a bisiminoether bridge with CuCl$_2$·2H$_2$O [20] (Figure 15). Very recently, Kokina et al. synthesized the mononuclear Cu(II) complex **71** and, through computational analysis, demonstrated that tricoordination is thermodynamically favored over dicoordination. The compound exhibited moderate cytotoxicity against the human breast adenocarcinoma cell line MCF-7 [33].

**Figure 15.** Reported copper compounds with iminoether containing ligands.

*3.6. Zinc and Cadmium Compounds*

There are a few examples of the synthesis and structural characterization of Zn(II) and Cd(II) complexes with chiral amino oxime ligands derived from natural terpenes, specifically from α-pinene, and the literature related was reviewed by Kokina et al. [14,109,110].

Enantioselective additions of alkynyl zinc reagents to both aromatic and aliphatic aldehydes have been explored starting from ZnEt$_2$, a variety of alkynes and chiral amino alcohol ligands derived from terpenes. Among the ligands tested, **30** (Figure 3) gave the best performance. Under these conditions, chiral propargylic alcohols were produced in good yields with moderate levels of enantioselectivity (Figure 16) [40].

**Figure 16.** Enantioselective alkynylations of aromatic and aliphatic aldehydes. Symbol * means stereogenic centers.

Glinskaya et al. published in 2012 the first structural study on a limonene-derived Zn(II) mononuclear derivative, compound **72** [111] (Figure 17). The molecule shows a pentadentate coordination environment around Zn(II), which adopts a distorted square–pyramidal geometry. Intramolecular hydrogen bonding between the oxime and oximato groups reinforces the stability of the compound and enhances its rigidity.

**Figure 17.** Zn(II) and Cd(II) compounds with limonene-derived ligands.

In contrast, the Zn(II) and Cd(II) complexes with tridentate amino bipyridine and O-picolyl α-amino ligands exhibit greater structural flexibility, with distorted geometries closer to trigonal bipyramidal [31–33]. The molecules displayed bright-blue luminescence, and the presence of terpene-derived ligands significantly enhanced this property. Compounds **75** and **76** were found to be fluorescent and active against the HepG2 liver cancer cell line, while ligand **10** showed negligible cytotoxicity (IC$_{50}$ > 50 μM). The cadmium complex **76** displayed an IC$_{50}$ value of 14.3 ± 1.5 μM, almost twice as high as that of cisplatin (IC$_{50}$ = 33.0 ± 5.4 μM).

## 4. Conclusions

A wide range of limonene-derived ligands have been stereoselectively synthesized for coordination with metal centers and as precursors in asymmetric catalysis, showing considerable promise in the enantioselective synthesis of metal complexes. When appropriately substituted, these chelating pro-ligands can yield enantiomerically pure compounds or promote diastereoselective mixtures of stereoisomers in solution. Thus, a significant number of limonene derivates, both free or coordinated to metal ions, have been synthesized and shown potential in a variety of enantioselective catalytic transformations including the transfer hydrogenation of ketones, allylation of β-diketones, and alkynylations of aromatic and aliphatic aldehydes, offering cost-effective alternative asymmetric catalytic systems.

The biological activity of chiral metal complexes based on limonene ligands has been less studied. However, it has been found that they exhibit not only promising antitumor properties against different cancer cell lines, but also significant antibacterial and antibiofilm properties against various pathogens, together with interesting luminescent properties. Further systematic research is needed to elucidate structure–property relationships and to develop innovative pharmaceutical drugs.

Looking forward, limonene-derived ligands hold promise for applications beyond traditional catalysis and medicinal chemistry. For example, their incorporation into photocatalytic systems could enable more efficient light-driven transformations, while their structural versatility may facilitate the design of unexplored bioconjugation strategies for targeted drug delivery. Additionally, their renewable and non-toxic nature positions them as attractive candidates for sustainable processes in industrial green chemistry, where cost-effective, environmentally friendly catalysts are increasingly in demand. Further exploration of these directions could expand both the synthetic and practical utility of limonene-based metal complexes.

**Funding:** This research was funded by Ministerio de Ciencia e Innovación (PID2019-108251RB18I00/AEI/10.13039/501100011033 and PID2023-148222OB-I00), and Universidad de Alcalá (UAH, Projects PIUAH22/CC-028, PIUAH23/CC-024).

**Data Availability Statement:** No new data were created or analyzed in this study.

**Conflicts of Interest:** The authors declare no conflicts of interest.

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
