# Peer review of "Chiral Transition Metal Complexes Featuring Limonene-Derived Ligands: Roles in Catalysis and Biology"

_inorganics, doi:10.3390/inorganics13100336_

Round 1
Reviewer 1 Report
Comments and Suggestions for Authors
In my opinion, the reviewed article is a valuable and carefully prepared review paper focused on transition-metal complexes with ligands derived from limonene. The authors present a broad overview of the field, covering synthetic methods, functionalization strategies, structural features, and applications of these systems in catalysis and biological contexts. I believe the topic has been chosen appropriately, as it aligns well with current trends in chemistry—particularly those involving the search for sustainable, naturally sourced chiral building blocks.
I am convinced that the diversity of described metals (Ti, Ru, Pd, Ni, Co, Zn, Cd, Cu) and the clear presentation of how structural modifications influence coordination behavior and catalytic activity are among the strongest aspects of this work. I think such a comprehensive overview can be useful not only to specialists in coordination chemistry but also to researchers interested in catalyst design or the development of biologically active metal complexes.
Despite its strengths, the article, in my view, requires several important revisions. First and foremost, the manuscript contains recurring editorial errors, such as multiple instances of “Error! Reference source not found”, which clearly result from citation software issues. I am convinced that thoroughly reviewing the reference list and standardizing the citation style would improve the article’s professionalism and readability.
I also noticed an imbalance in the level of detail between different sections. While the parts dedicated to ruthenium and palladium are very detailed, the sections covering nickel, cobalt, copper, or zinc are far more superficial. In my opinion, the less-developed parts should be expanded or at least supplemented with additional literature examples and structural context. Consistency in citation style—whether numerical or author-date—would also enhance the article’s clarity.
I believe that the language and style of the manuscript could benefit from simplification. Many sentences are overly long and syntactically dense, which may hinder comprehension—especially for readers outside the immediate field. In my view, shortening and restructuring sentences, removing unnecessary repetitions, and improving clarity would significantly enhance the quality of the writing.
I also find that the biological data presented in the article could be more clearly organized. Although results related to anticancer and antimicrobial activity are included, they are scattered throughout the text. I am convinced that adding a summary table with ICâ‚…â‚€ values, tested cell lines, and comparisons to standard drugs such as cisplatin would improve accessibility. Additionally, I think it would be valuable to expand the discussion of proposed biological mechanisms, including DNA interaction and cytotoxicity pathways.
From my perspective, the inclusion of visual summaries—such as tables compiling catalytic performance or simplified reaction mechanisms (e.g., ruthenium-catalyzed transfer hydrogenation)—would further strengthen the review. I believe such additions would help readers synthesize information and make the article more user-friendly.
Finally, I would like to stress that, in my opinion, the conclusion of the article should be improved. While it provides a summary of the main findings, it lacks a forward-looking perspective. I think it would be beneficial to add a paragraph outlining future research directions and practical applications—for example, the use of limonene-derived ligands in photocatalysis, medicinal chemistry, bioconjugation strategies, or industrial green chemistry.
To summarize, I am convinced that this article has strong potential and presents an interesting and timely contribution to the field of coordination chemistry. The authors address a relevant topic with broad implications for bioinorganic chemistry and catalysis. However, I believe that before publication, some editorial and content-related improvements are necessary. Once these revisions are implemented, the article may serve as a valuable and frequently cited reference for future research in this rapidly developing area.
Author Response
Comment 1: In my opinion, the reviewed article is a valuable and carefully prepared review paper focused on transition-metal complexes with ligands derived from limonene. The authors present a broad overview of the field, covering synthetic methods, functionalization strategies, structural features, and applications of these systems in catalysis and biological contexts. I believe the topic has been chosen appropriately, as it aligns well with current trends in chemistry—particularly those involving the search for sustainable, naturally sourced chiral building blocks.
I am convinced that the diversity of described metals (Ti, Ru, Pd, Ni, Co, Zn, Cd, Cu) and the clear presentation of how structural modifications influence coordination behavior and catalytic activity are among the strongest aspects of this work. I think such a comprehensive overview can be useful not only to specialists in coordination chemistry but also to researchers interested in catalyst design or the development of biologically active metal complexes.
Despite its strengths, the article, in my view, requires several important revisions. First and foremost, the manuscript contains recurring editorial errors, such as multiple instances of “Error! Reference source not found”, which clearly result from citation software issues. I am convinced that thoroughly reviewing the reference list and standardizing the citation style would improve the article’s professionalism and readability.
Response 1: I do not understand the origin of these errors. As I have confirmed now, all given DOI’s link to the appropriate article online, the style is consistent with the recommendations given in the Inorganics web page and, to the best of my knowledge, there were only minor editing mistakes. I encourage the reviewer to revise the pdf version of the manuscript, instead of the word file which includes the Mendeley fields (software used for the styling of references). Maybe the software is the problem.
Comment 2: I also noticed an imbalance in the level of detail between different sections. While the parts dedicated to ruthenium and palladium are very detailed, the sections covering nickel, cobalt, copper, or zinc are far more superficial. In my opinion, the less-developed parts should be expanded or at least supplemented with additional literature examples and structural context. Consistency in citation style—whether numerical or author-date—would also enhance the article’s clarity.
Response 2: The imbalance noted by the reviewer reflects the fact that there is more scientific knowledge on Pd and Ru compounds containing limonene than on those of other metals. I would like to highlight that Pd–terpene compounds were among the first to be prepared, as early as 1970. Regarding the citation style, I am uncertain about the concern raised; in the version I have, the style is consistent, and we have always used numerical citations. Nevertheless, I have carefully reviewed all references again.
Comment 3: I believe that the language and style of the manuscript could benefit from simplification. Many sentences are overly long and syntactically dense, which may hinder comprehension—especially for readers outside the immediate field. In my view, shortening and restructuring sentences, removing unnecessary repetitions, and improving clarity would significantly enhance the quality of the writing.
Response 3: I thank the reviewer for the comment. However, without more specific details, I find the observations on my writing style to be somewhat subjective, and I am unsure how I could improve it further to meet expectations. Again, we have carefully read the manuscript, and some modifications have been made. They are listed below:
- Abstract: The order of sentences has been modified to gain clarity.
- Lines 63, 69, 80, on page 2.
- Lines 81, 106, on page 3.
- Lines 124-126, on page 4
- Lines 158, 163, 174, on page 5
- Lines 200-201, on page 6
- Captions of figure 7, 12 and 13
- Lines 287, 288, 304 on page 8
- Lines 317-323, 349, on page 9
- Lines 350, page 10.
- Lines 473-477 and 485-486 on page 13.
- Lines 492, 498, 501-502, 505, 506, on page 14.
Comment 4: I also find that the biological data presented in the article could be more clearly organized. Although results related to anticancer and antimicrobial activity are included, they are scattered throughout the text. I am convinced that adding a summary table with ICâ‚…â‚€ values, tested cell lines, and comparisons to standard drugs such as cisplatin would improve accessibility. Additionally, I think it would be valuable to expand the discussion of proposed biological mechanisms, including DNA interaction and cytotoxicity pathways.
From my perspective, the inclusion of visual summaries—such as tables compiling catalytic performance or simplified reaction mechanisms (e.g., ruthenium-catalyzed transfer hydrogenation)—would further strengthen the review. I believe such additions would help readers synthesize information and make the article more user-friendly.
Response 4: We appreciate the reviewer’s observation. However, the aim of this minireview is to highlight the most recent advances in the field, rather than to provide a comprehensive survey of all available data. Our intention was to emphasize emerging trends and key findings, while keeping the scope focused and concise.
To clarify this point we have added a sentence in the introduction, please, see lines 46-47 on page 2.
Regarding DNA interactions, Lines 298-301, page 8 and 431-434, page 12 have been added.
Comment 5: Finally, I would like to stress that, in my opinion, the conclusion of the article should be improved. While it provides a summary of the main findings, it lacks a forward-looking perspective. I think it would be beneficial to add a paragraph outlining future research directions and practical applications—for example, the use of limonene-derived ligands in photocatalysis, medicinal chemistry, bioconjugation strategies, or industrial green chemistry.
Response 5: We appreciate this reviewer’s comment. A correction and an additional paragraph (please, see lines 495, 498-506 on page 14) have been added to include the potential future perspectives of such compounds.
Reviewer 2 Report
Comments and Suggestions for Authors
Amrani and coworkers submitted a manuscript entitled “Chiral Transition-Metal Complexes Featuring limonene-derived Ligands: Roles in Catalysis and Biology”. This manuscript provides a comprehensive review of recent advances in the synthesis, the structural properties, and reactivity of chiral metal complexes incorporating limonene-derived chiral ligands. It serves as a valuable contribution to the field of chiral pool utilization. The manuscript is suitable for publication after addressing the following issues:
- The introduction primarily discusses terpenes, but the focus of the manuscript is on limonene-derived ligands. Please explicitly clarify the relationship between terpenes and limonene to improve coherence.
- The is a duplicated figure 3 in page 4.
- On line 219 Page 7, the X-ray analysis shouldpresumably refer to compound 39 instead of 59. Please confirm and revise
- Figure 6, the caption is inaccurate, as 38 is an iron complex.
- The word "symmetry" is misspelled in Figure 8.
- Some references appear to contain errors in the text. Please carefully review and ensure all references are accurate and properly cited.
Author Response
Amrani and coworkers submitted a manuscript entitled “Chiral Transition-Metal Complexes Featuring limonene-derived Ligands: Roles in Catalysis and Biology”. This manuscript provides a comprehensive review of recent advances in the synthesis, the structural properties, and reactivity of chiral metal complexes incorporating limonene-derived chiral ligands. It serves as a valuable contribution to the field of chiral pool utilization. The manuscript is suitable for publication after addressing the following issues:
Comment 1:
- The introduction primarily discusses terpenes, but the focus of the manuscript is on limonene-derived ligands. Please explicitly clarify the relationship between terpenes and limonene to improve coherence.
Response 1: According to the reviewer’s suggestion, we have added a brief explanation on this matter (see Lines 30-33 Page 1, and changes to line 50, page 2.
Comment 2:
- The is a duplicated figure 3 in page 4.
Response 2: We thank the reviewer for highlighting the mistake which has been corrected.
Comment 3:
- On line 219 Page 7, the X-ray analysis shouldpresumably refer to compound 39 instead of 59. Please confirm and revise
Response 3: We thank the reviewer for this observation. The mistake has been corrected, the compound was, indeed, 39.
Comment 4:
- Figure 6, the caption is inaccurate, as 38 is an iron complex.
Response 4: We thank the reviewer for this observation. The caption of Figure 6 has been modified to include Fe-derivative.
Comment 5:
- The word "symmetry" is misspelled in Figure 8.
Response 5: We thank the reviewer for this observation. The misspelling has been corrected.
Comment 6:
- Some references appear to contain errors in the text. Please carefully review and ensure all references are accurate and properly cited.
Response 6: All references have been carefully reviewed and corrected when needed.
Reviewer 3 Report
Comments and Suggestions for Authors
The present manuscript by Royo, El Amrani and coworkers is already at its second revision. Thus, I will keep my comments to the minimum. The work is worth to be published in Inorganics. Some sentences are still a bit complex to follow, but the meaning is anyway clear. However, there are some points that need to be corrected before the manuscript can be accepted for publication:
Page 2, line 70. "chloride" should be "chloro"
Page 3, lines 91,92 "other green-solvents based in alcohols" means nothing. Rephrase.
Page 3, line 93. "ammines" should be "amines"
Page 4, line 127: " alcohol phospane" should be "phosphinalcohol"
Page 4 line 137 " have shown interesting luminescent properties." seems to imply that all ligands in the previous sentence are luminescent, whereas they are only the Zn(II) and Cd(II) complexes of 35 to show this property. Split the sentence and rephrase it to make this clear.
Page 4 Figure 4: in compound 35, the group indicated as -NH(OAc) should be –NH(Ac)
Page 6, line 196: "…. DU-145, some of them, specially 37," " DU-145. Some of them, especially 37,"
Page 6, line 217. Ref 63 is dated 1984, not 1985
Page 7, line 260. I guess the relevant complexes are 41 and 42, not 40 and 41
Page 8, line 275: "those" should be "these"
Page 8, line 313: "demonstrated" should be "showed"
Several references are not correctly reported.
I don't know why the authors changed "Dalton Trans." into "Dalt. Trans.". The former was correct and must be reintroduced in all cases in which it has been changed.
ref 28 and 48: "Inorganica" must be "Inorg."
ref 69 "European" must be "Eur."
Author Response
COMMENT 1: The present manuscript by Royo, El Amrani and coworkers is already at its second revision. Thus, I will keep my comments to the minimum. The work is worth to be published in Inorganics. Some sentences are still a bit complex to follow, but the meaning is anyway clear. However, there are some points that need to be corrected before the manuscript can be accepted for publication:
Page 2, line 70. "chloride" should be "chloro"
Page 3, lines 91,92 "other green-solvents based in alcohols" means nothing. Rephrase.
Page 3, line 93. "ammines" should be "amines"
Page 4, line 127: " alcohol phospane" should be "phosphinalcohol":
Page 4 line 137 " have shown interesting luminescent properties." seems to imply that all ligands in the previous sentence are luminescent, whereas they are only the Zn(II) and Cd(II) complexes of 35 to show this property. Split the sentence and rephrase it to make this clear.
Page 4 Figure 4: in compound 35, the group indicated as -NH(OAc) should be –NH(Ac)
Page 6, line 196: "…. DU-145, some of them, specially 37," " DU-145. Some of them, especially 37,"
Page 6, line 217. Ref 63 is dated 1984, not 1985
Page 7, line 260. I guess the relevant complexes are 41 and 42, not 40 and 41
Page 8, line 275: "those" should be "these"
Page 8, line 313: "demonstrated" should be "showed"
Several references are not correctly reported.
I don't know why the authors changed "Dalton Trans." into "Dalt. Trans.". The former was correct and must be reintroduced in all cases in which it has been changed.
ref 28 and 48: "Inorganica" must be "Inorg."
ref 69 "European" must be "Eur."
RESPONSE 1: We would like to thank the reviewer for the careful revision, which has significantly improved our manuscript. All the corrections have been made as suggested.
Reviewer 4 Report
Comments and Suggestions for Authors
In this review, the authors focus on the application of chiral transition-metal complexes functionalized with tailored-designed limonene ligands. This work is nicely conceived however it is really difficult to follow the application in catalysis of such catalysts, without showing a schemes the synthetic transformation. I think that this work can become suitable for publication but necessitates a strong implementation of the schemes proposed. I strongly suggest to draw schemes highlighting the synthetic transformation.
Minor issue - I found a bit measileading this statement in line 367 "Palladium plays a prominent role in organometallic chemistry, providing a crucial catalytic platform for a broad range of reactions, such as the Heck, Suzuki and Suzuki-Miyaura cross-couplings". In fact, palladium is the most powerful catalyst in Nature and has been explored for many other outstanding catalytic reactions and particularly, C-C bond forming reactions such as Tsuji-Trost allylations and C-H activations. I would implement this part. For leading reviews, you might cite: Angew. Chem. Int. Ed. 2024, 63, e202314637; ChemCatChem, 2022, 14, e202200295; Chem. Rev. 2017, 117, 8754–8786; https://doi.org/10.1016/j.trechm.2021.12.005.
Author Response
COMMENT 1: In this review, the authors focus on the application of chiral transition-metal complexes functionalized with tailored-designed limonene ligands. This work is nicely conceived however it is really difficult to follow the application in catalysis of such catalysts, without showing a schemes the synthetic transformation. I think that this work can become suitable for publication but necessitates a strong implementation of the schemes proposed. I strongly suggest to draw schemes highlighting the synthetic transformation.
RESPONSE 1: We would like to thank the reviewer for the comments, which have improved the quality of the manuscript.
To gain clarity, figures 9, 13 and 15 have been added. A paragraph regarding Figure 15 has been included, since its omission was a mistake.
COMMENT 2: Minor issue - I found a bit measileading this statement in line 367 "Palladium plays a prominent role in organometallic chemistry, providing a crucial catalytic platform for a broad range of reactions, such as the Heck, Suzuki and Suzuki-Miyaura cross-couplings". In fact, palladium is the most powerful catalyst in Nature and has been explored for many other outstanding catalytic reactions and particularly, C-C bond forming reactions such as Tsuji-Trost allylations and C-H activations. I would implement this part. For leading reviews, you might cite: Angew. Chem. Int. Ed. 2024, 63, e202314637; ChemCatChem, 2022, 14, e202200295; Chem. Rev. 2017, 117, 8754–8786; https://doi.org/10.1016/j.trechm.2021.12.005.
RESPONSE 2: We have added the references suggested by the reviewer and rewritten the sentence to include them.
Round 2
Reviewer 2 Report
Comments and Suggestions for Authors
Amrani and coworkers have submitted a revised version of their manuscript “Chiral Transition-Metal Complexes Featuring limonene-derived Ligands: Roles in Catalysis and Biology”. The modifications suggested the reviewer have been done seriously, leading to a significant improvement in the manuscript’s quality. Accordingly, I feel that the manuscript now meets the publication standards of Inorganics and recommend its acceptance.
Author Response
COMMENT: Amrani and coworkers have submitted a revised version of their manuscript “Chiral Transition-Metal Complexes Featuring limonene-derived Ligands: Roles in Catalysis and Biology”. The modifications suggested the reviewer have been done seriously, leading to a significant improvement in the manuscript’s quality. Accordingly, I feel that the manuscript now meets the publication standards of Inorganics and recommend its acceptance.
RESPONSE: We would like to thank the reviewer for the first and second revision, which have significantly increased the quality of the manuscript.
Reviewer 4 Report
Comments and Suggestions for Authors
The authors revised the manuscript accordingly to this reviewer's suggestion. I think that the work has been improved and represents a nice overview on the organometallic properties of TM complexes featuring limonene-based ligands. Before to recommend its acceptance, I have to ask the authors to amend Figures 13 and 16. The geometries of C-C bonds of the molecular structures are incorrect. This is not convenient when you want to publish a paper in a chemistry journal. If you used ChemDraw to draw the structures, I suggest to use the "clean-up structure" function. Other than that, I think that this paper is well-written, clear to the reader and deserves to be published in Inorganics
Author Response
Comment 1: The authors revised the manuscript accordingly to this reviewer's suggestion. I think that the work has been improved and represents a nice overview on the organometallic properties of TM complexes featuring limonene-based ligands. Before to recommend its acceptance, I have to ask the authors to amend Figures 13 and 16. The geometries of C-C bonds of the molecular structures are incorrect. This is not convenient when you want to publish a paper in a chemistry journal. If you used ChemDraw to draw the structures, I suggest to use the "clean-up structure" function. Other than that, I think that this paper is well-written, clear to the reader and deserves to be published in Inorganics
Response:
Figures 13 and 16 have been corrected. However, I would like to highlight that absolute configurations are intentionally omitted, as the purpose is to depict a general scheme for a series of enantioselective transformations.